# Full HAWT rotor CFD simulations using different RANS turbulence models compared with actuator disk and experimental measurements

Nikolaos Stergiannis<sup>1,2,3</sup>, Jeroen van Beeck<sup>2</sup>, and Mark C. Runacres<sup>1</sup>

 <sup>1</sup>Vrije Universiteit Brussel, Faculty of Engineering, Thermo and Fluid Dynamics Research Group (FLOW), Pleinlaan 2, B-1050, Brussels, Belgium
 <sup>2</sup>Von Karman Institute for Fluid Dynamics, Department of Environmental and Applied Fluid Dynamics, Waterloosesteenweg 72, B-1640, Brussels, Belgium
 <sup>3</sup>3E S.A., Kalkkaai 6, B-1000, Brussels, Belgium

Correspondence to: Nikolaos Stergiannis (nikolaos.stergiannis@vub.ac.be)

## Abstract.

The development of large-scale wind energy projects has created the demand for increasingly accurate and efficient models that limit a project's uncertainties and risk. Wake effects are of great importance and are relevant for the optimization of wind farms. Despite a growing body of research, there are still many open questions and challenges to overcome. In computational

- modelling, there are always numerous input parameters such as material properties, geometry, boundary conditions, initial conditions, turbulence modelling etc. whose estimation is difficult and their values are often inaccurate or uncertain. Due to the lack of information of several sources, e.g., uncertainties present in operating conditions as well as in the mathematical modelling, the computational output is also uncertain. It is therefore very important to validate the mathematical models with experiments performed in controlled conditions. In the present paper, the single wake characteristics of a Horizontal-Axis
- Wind Turbine Rotor (HAWT) and their spatial evolution are investigated with different Computational Fluid Dynamics (CFD) modelling approaches and compared to experimental measurements.

The steady state 3-D Reynolds-Averaged Navier Stokes (RANS) equations are solved in the open-source platform Open-FOAM, using different turbulence closure schemes. For the full-rotor CFD simulations, the Multiple Reference Frames (MRF) approach was used to model the rotation of the blades. For the simplified cases, an actuator disk model was used with the exper-

- imentally measured thrust  $(C_T)$  and power  $(C_P)$  coefficient values. The performance of each modelling approach is compared with experimental wind tunnel wake measurements from the 4th blind test organized by NOWITECH and NORCOWE in 2015. Numerical results are compared with experimental data along three horizontal lines downstream, covering all the wake regions. Wake predictions are shown to be very sensitive to the choice of the RANS turbulence model. For most cases, the ADM under-predicts the velocity deficit, except for the case of *RNG* k- $\varepsilon$  which showed a superb performance in the mid and
- far wake. The full wind turbine rotor simulations showed good agreement to the experimental data, mainly in the near wake, amplifying the differences between the simplified models.

## 1 Introduction

During the lifetime of a wind farm, the operating wind turbines have to deal with multiple interactions. Part of the wind's kinetic energy is extracted by the rotors, resulting in a velocity deficit and increased levels of turbulence downstream (Vermeer et al., 2003). Wake effects can cause total annual power losses up to 30%.

- The wind turbines in a row are exposed to single or multiple upstream wake flows, resulting in power losses and additional loads from increased turbulence fluctuations (Rados et al., 2012; Prospathopoulos et al., 2011; Barthelmie et al., 2007; Larsen et al., 1996, 1998, 2003; Magnusson et al., 1996). To optimize a wind farm over a certain spatial area and reach the maximum potential production of the installed wind turbines, we have to understand the physics involved in the process of making torque from wind and be able to predict effectively the wake effects. The estimation of the wake characteristics helps the developers to
- optimize the final wind farm layout under the prevailing winds, ensuring improved power extraction and working conditions for the downstream wind turbine generators. Wake optimization can benefit the project's lifetime OPerational EXpenses (OPEX), since operating wind turbine generators are expected to be more reliable with less total failures (Larsen et al., 1998).

To predict wakes, several linear models have been developed and applied in industry (Jensen, 1983; Katic et al., 1986; Ainslie, 1988; Rados et al., 2002). However, despite their ease of use, all these models are based on simple assumptions and

- experimental observations. On the other hand, Computational Fluid Dynamics (CFD) simulations provide more sophisticated methods, such as the 3-dimensional solution of the Navier-Stokes equations, that represent the flow field with great accuracy (Mikkelsen, 2003). Although CFD simulations can be performed on both steady and unsteady state problems, it is common and acceptable to use, with certain limitations, the assumption of steady state approach to resolve a time-varying (unsteady state) problem when one is interested in the mean values of the flow. The numerical solution of the RANS equations using a two-
- equation turbulence closure model (Markatos, 1986; Wilcox, 1993) is a common approach with reasonable computational cost for industrial applications involving flows of high Reynolds numbers. Methods that are inherently unsteady, such as Detached Eddy Simulation (DES), Large Eddy Simulation (LES) and Direct Numerical Simulation (DNS) can be more accurate but the high computational cost makes them less suitable for use in industrial applications. However, following the technological growth, the computational cost is expected to decrease in years to come and the usage of CFD will be increasingly affordable
- for optimizing complex industrial applications.

## 2 Mathematical model

#### 2.1 Statement of the problem

The physical problem under investigation is the flow over two identical wind turbines in a row under the controlled conditions of a wind tunnel. The current study focuses on predicting the single wake characteristics and development using different CFD modelling approaches.

5

# 2.2 Governing equations

The governing equations solved of the flow field are the continuity and conservation of momentum equations:

$$\frac{\partial \rho}{\partial t} + \nabla \cdot (\rho \boldsymbol{u}) = 0$$
(1)
$$\frac{\partial (\rho \boldsymbol{u})}{\partial t} + \boldsymbol{u} \cdot \nabla (\rho \boldsymbol{u}) = -\nabla p + \nabla \cdot \tau + \boldsymbol{S}_{\boldsymbol{M}}$$
(2)

where  $\rho$  is the air density,  $\boldsymbol{u}$  is the fluid velocity vector, p the pressure,  $\tau$  the shear stress tensor and  $\boldsymbol{S}_{\boldsymbol{M}}$  a momentum source term. The stress tensor is related to the strain rate as:

$$\tau = \mu \left[ \nabla \boldsymbol{u} + \left( \nabla \boldsymbol{u} \right)^{\mathrm{T}} - \frac{2}{3} \cdot \boldsymbol{\delta} \cdot \nabla \boldsymbol{u} \right] = 0$$
(3)

where  $\delta$  is Kronecker's delta and  $\mu$  is the dynamic viscosity of the fluid.

## 10 2.3 The Multiple Reference Frame (MRF) formulation

The multiple reference frame (MRF) approach was used to model the internal rotating frames in a stationary computational mesh and reference frame. The velocities in the inertial and rotating reference frames, using the notation I for inertial and R for rotating, are related by:

$$\boldsymbol{u}_{\mathrm{I}} = \boldsymbol{u}_{\mathrm{R}} + \boldsymbol{\Omega} \times \boldsymbol{r} \tag{4}$$

15 The acceleration is expressed as:

$$\left[\frac{d\boldsymbol{u}_{\mathrm{I}}}{dt}\right]_{\mathrm{I}} = \left[\frac{d\boldsymbol{u}_{\mathrm{R}}}{dt}\right]_{\mathrm{R}} + \frac{d\boldsymbol{\Omega}}{dt} \times \boldsymbol{r} + 2\boldsymbol{\Omega} \times \boldsymbol{u}_{\mathrm{R}} + \boldsymbol{\Omega} \times \boldsymbol{\Omega} \times \boldsymbol{r}$$
(5)

The incompressible RANS equations for steady flow in the rotating frame of reference can be written in terms of relative velocity:

$$\nabla \cdot \boldsymbol{u}_{\mathrm{R}} = 0 \tag{6}$$

20 
$$\nabla \cdot (\boldsymbol{u}_{\mathrm{R}} \otimes \boldsymbol{u}_{\mathrm{R}}) + 2\boldsymbol{\Omega} \times \boldsymbol{u}_{\mathrm{R}} + \boldsymbol{\Omega} \times \boldsymbol{\Omega} \times \boldsymbol{r} = -\nabla \left(\frac{p}{\rho}\right) + \nu \nabla \cdot \nabla (\boldsymbol{u}_{\mathrm{R}})$$
 (7)

#### 2.4 Turbulence modelling

Two-equation turbulence models have been widely tested over the years and have been proven as an accepted compromise between accuracy and computational cost. In the present work, the performance of several two-equation turbulence models on wake expansion is tested and compared with measurements.

## 25 2.4.1 Standard k- $\varepsilon$

The standard k- $\varepsilon$  turbulence model (Launder and Spalding, 1974) is based on transport equations for the turbulence kinetic energy, k and its dissipation rate,  $\varepsilon$ . The model parameters used are given in Table 1, following Launder et al (Launder et al., 1973).

**Table 1.** Constants of the *standard* k- $\varepsilon$  turbulence model

| $C_{\mu}$ | $C_1$ | $C_2$ | $\sigma_k$ | $\sigma_{\varepsilon}$ |
|-----------|-------|-------|------------|------------------------|
| 0.09      | 1.44  | 1.92  | 1.0        | 1.3                    |

## 2.4.2 Realizable k- $\varepsilon$

In the *realizable* k- $\varepsilon$  turbulence model of Tsan-Hsing Shih et al, a modified model dissipation rate equation and realizable eddy viscosity formulation is used. A detailed description of the turbulence model is given in (Shih et al., 1995). The model 5 constants are given in Table 2.

**Table 2.** Constants of the *realizable* k- $\varepsilon$  turbulence model

| $C_{\mu}$ | $A_0$ | $C_2$ | $\sigma_k$ | $\sigma_{\varepsilon}$ |
|-----------|-------|-------|------------|------------------------|
| 0.09      | 4.0   | 1.9   | 1.0        | 1.3                    |

## 2.4.3 RNG k-ε

The RNG k- $\varepsilon$  turbulence model was developed by Yakhot et al. (Yakhot et al., 1992) using the Re-Normalisation Group (RNG) methods in the Navier-Stokes equations in order to account for smaller scales of turbulence. The main difference and advantage

10

compared to the *standard* k- $\varepsilon$  turbulence model, is that the model attempts to account for the different scales of motion through a modified epsilon equation, instead of determining the eddy viscosity from a single turbulence length scale. The model constants are given in Table 3.

**Table 3.** Constants of the *RNG* k- $\varepsilon$  turbulence model

| $C_{\mu}$ | $C_1$ | $C_2$ | $\sigma_k$ | $\sigma_{arepsilon}$ | $\eta_0$ | $\beta$ |
|-----------|-------|-------|------------|----------------------|----------|---------|
| 0.0845    | 1.42  | 1.68  | 0.71942    | 0.71942              | 4.38     | 0.012   |

#### 2.4.4 Wilcox (1988) k- $\omega$

15

The k- $\omega$  turbulence model is a two-equation turbulence model that uses the turbulence kinetic energy k and the specific rate of dissipation  $\omega$  to predict turbulence. The model has been improved over the years with several versions and modifications (Wilcox, 1993, 2008), but in our cases we are using the Wilcox (1988) version (Wilcox, 1988) that is implemented in Open-FOAM ((OpenFOAM, 2016)). The model constants are given in Table 4.

**Table 4.** Constants of the *standard* k- $\omega$  turbulence model

| $C_{\mu}$ | $\sigma_k$ | $\sigma_{\omega}$ | $\beta^*$ | $\beta$ | $\gamma$ |
|-----------|------------|-------------------|-----------|---------|----------|
| 0.09      | 0.5        | 0.5               | 0.09      | 0.072   | 0.52     |

#### 2.4.5 Menter (2003) k-ω SST

The k-ω SST turbulence model, also known as Menter's Shear Stress Transport turbulence model, was introduced in 1994 by
F.R. Menter (Menter, 1994). It is a two-equation turbulence model that uses k-ω in the inner region of the boundary layer and switches to k-ε in the free shear flow to improve the predictions of adverse pressure gradients. The model also has other modified versions (Hellsten, 1998). The updated model of 2003 (Menter et al., 2003) with a different expression to derive the eddy viscosity was used. The model constants are given in Table 5.

**Table 5.** Constants of  $k - \omega SST$  turbulence model

| $\sigma_{k1}$ | $\sigma_{k2}$ | $\sigma_{\omega 1}$ | $\sigma_{\omega 2}$ | $\beta^*$ | $\beta_1$ | $\beta_2$ | $\gamma_1$ | $\gamma_2$ | $\alpha_1$ |
|---------------|---------------|---------------------|---------------------|-----------|-----------|-----------|------------|------------|------------|
| 0.85          | 1.0           | 0.5                 | 0.856               | 0.09      | 0.075     | 0.0828    | 5/9        | 0.44       | 0.31       |

#### 2.5 Wind turbine modelling

10 In the case of actuator disk model, wind turbine rotors are approximated as momentum sinks (see eq. 2) to represent the axial thrust force T, associated with a constant uniform thrust coefficient  $C_T$  over the rotor area:

$$T = 0.5\rho C_T A U_{\rm ref}^2 \tag{8}$$

where A is the surface area of the rotor-disk,  $\rho$  the air density,  $U_{ref}$  the undisturbed reference free-stream velocity and  $C_T$  the thrust coefficient of the rotor.

- Once the  $U_{ref}$  is known,  $C_T$  can be estimated through the thrust curve of the wind turbine generator considered as uniform over the rotor area (Rados et al., 2012; Crasto et al., 2012; Mikkelsen, 2003). However, in operating conditions, the flow across the rotor is very complex with varying span-wise properties, due to the blade characteristics, rotational velocity, turbulence, finite number of blades and also other flow characteristics related to non-uniform inflow conditions, atmospheric boundary layer shear and so on.
- To overcome the limitations of the standard actuator disk model, several more advanced models like the generalized actuator disk, actuator line model, actuator surface, have been proposed (Mikkelsen, 2003). The drawback of the advanced models is mainly that they need several input variables such as airfoil data, detailed blade geometry which most of the times are not available or confidential for industrial applications. Finally, most of the advance models require time resolved CFD simulations which increase the final computational cost (Sanderse et al., 2011).

5

25

The standard actuator disk model that was used in this study is implemented in OpenFOAM based on the axial induction factor *a*:

$$a = 1 - \frac{C_P}{C_T} \tag{9}$$

where  $C_P$  is the power coefficient. The thrust force can then be expressed as:

$$T = 2\rho Aa(1-a)U_{\rm ref}^2 \tag{10}$$

For all the cases under investigation, a constant uniform reference wind speed of 11.5 m/s has been used. The coefficients 10  $C_P$  and  $C_T$  have been provided from the wind tunnel measurements and were used as inputs to the simplified models.

#### 3 The 4th blind test experiment

The 4th Blind Test (BT4) experiment was organized by NOWITECH and NORCOWE in 2015. The total power output from two in-line turbines was investigated under the influence of different inlet conditions and turbine separation distance (Sætran and Bartl, 2015).

- 15 The axial separation distance between the turbines was set to x/D = 2.77, x/D = 5.18 and x/D = 9.00, where the diameter D = 0.894 m. Furthermore, three different inflow conditions at the inlet of the test section were tested:
  - Low-turbulence uniform inflow (CASE-A): No grid at the inlet to the test section. At the position of the upstream turbine the turbulence intensity measured was TI = 0.23%. The mean wind speed was uniform across the test section, apart from small wall boundary layer effects.
- High-turbulence uniform inflow (CASE-B): An evenly spaced grid at the tunnel inlet generated a higher turbulence intensity level of TI = 10.0% at the location of the upstream turbine. The mean wind speed was uniform across the test section.
  - High-turbulence shear inflow (CASE-C): A turbulence grid with increasing vertical distance between the horizontal bars was installed at the inlet of the test section, creating a non-uniform shear flow with a mean turbulence intensity of TI = 10.0% over the rotor swept area of the upstream turbine.

At the experiment, the high turbulence intensity CASE-B was tested on three different downstream axial separation distances. In our study, the CASE-B3 of high TI and with axial separation distance of x/D = 9.00 was chosen, mainly because more measured data were available downstream and additionally to eliminate the impact of the second wind turbine on the single wake expansion of the first turbine. For the simulations, a uniform inlet velocity profile of 11.5 m/s was considered, similar to the inlet velocity profile of the wind tunnel. The two wind turbines are identical and constructed with the same aluminium blades, using the NREL S826 airfoil section from root to tip.

# 5 4 Full rotor CFD simulations

To address the question of how much information is lost with the simplified models, results are compared with more advanced CFD simulations that include the full wind turbine rotor geometries and their hubs.

## 4.1 Computational domain

The computational domains were designed to match the exact wind tunnel dimensions to represent the experimental setup and account for possible blockage effects (Fig. 1). Particularly for the CASE-B3, the domain have been extended by 3.7 m in length (30% of total) to avoid any numerical oscillations from the outlet at the far wake measurements downstream.