# Peer review of "Full HAWT rotor CFD simulations using different RANS turbulence models compared with actuator disk and experimental measurements"

_Wind Energy Science, 2017_

## Referee Comment (RC1) · Anonymous Referee #1 · 20 Mar 2017

The authors present a study of RANS simulations of the double rotor experiment of NTNU. Results are presented using blade-resolved simulations in a rotating reference frame, as well as using an ADM. Four different eddy-viscosity models are compared. Unfortunately, I cannot recommend this manuscript for publication. First of all, the authors fail to provide a decent literature survey on blade-resolved simulations of wind turbines. In fact, they do not cite a single reference (making it appear as if they are the first to do blade resolved simulations), though current state-of-the art include, e.g., adjoint-based aeroelastic rotor resolved optimization (see, the work of Mavripilis at Wyoming – the same group also performs wind-farm simulation using full rotor models),

the work of Bazilevs (2011; full aeroelastic rotor simulations), the work in Stuttgart on blade resolved DES, just to name a few. Moreover, given the current state of the art in this field, I fail to see what the innovation is in the current manuscript. The authors do not identify this: there is no real research question or hypothesis, and consequently no relevant conclusions follow. Moreover, the one conclusion they formulate (k-omega models are the best), is not really substantiated as they do not include tower or nacelle in their simulations. Finally, there are many inaccuracies, and trivial statements – see further below for some further details.

Some more detailed issues

- page 1, line 21: don't understand that phrase...

- page 2, line 3: 'wake effects can cause total annual power losses up to 30%' What wind farm are you talking about; give reference and details. Seems rather big (unless maybe extreme case such as Lillgrund)

- page 2, line 18: 'it is common and acceptable to use, with certain limitations, the assumption of steady state approach to resolve a time-varying (unsteady state) problem when one is interested in the mean values of the flow.' What do you mean by that? What limitations? Please substantiate

- Eqs 1 and 2: these are not the correct RANS equations

- Section 2.4: belongs in appendix at best

- page 5, line 18: 'However, in operating conditions, the flow across the rotor is very complex, ..., and also other flow characteristics related to non-uniform inflow conditions, atmospheric boundary layer and so on'. True, but last situations also do not occur in you blade-resolved simulations (where you use uniform inflow); moreover, they could be accounted for in ADM, e.g., the rotating ADM by the group of Porté-Agel.

- page 10, line 5: '... approximately 6000 iterations and were run for 10000 iterations'. Really?? And that for a residual reduction of only 4 orders of magnitude?? At this

number of iterations, why not do DES – these would typically require between 10000 and 100000 time steps?

- connected to previous point: what is the computational time, number of processors, possible parallel efficiency, etc.

- page 10, line 7: 'Full rotor simulations are capable to represent the flow field close to the wind turbine (Fig. 3), but they are over-estimating the velocity deficit at the far wake (Fig. 9)' Why? No real explanation seems to be given in the paper

- number and order figures according to occurrence in the text

- page 11, line 12: 'are in agreement with the contours of vorticity (Fig. 7a) in which the strong effect of the shear stresses at the edge of the disk area is apparent' I expect the effect of shear stresses only to become apparent more downstream – near the disk, solution will behave more like potential flow.

- page 11, line 16: what recirculation zone – do you find a recirculation zone near the ADM? Did you correctly implement the model?

- page 11, line 18: '. . . is a strong connection between the production of turbulence and the wake recovery' Sorry, but this is really a rather trivial statement

- page 15, line 8: '. . ., we expect equal production of turbulence from the two rotors'. How so? Can you explain why you expect that? Not clear to me, since shear as well as turbulence levels of at second turbine are different, so also production will be different

- why not show comparison with experimental results of the second wake? Why also not consider the closer spaced cases?

- page 18: it is concluded that the k-omega models are best. Unfortunately, no tower or nacelle is modelled, so not sure that this conclusion will hold once you would add them

- page 18, line 17: '. . . ADM was observed also by other studies (. . ., Sanderse et al

2011, . . .)' Sanderse et al is a review paper and not an ADM study

---

## Referee Comment (RC2) · Anonymous Referee #2 · 27 Mar 2017

The paper addresses the flow around two identical wind turbines placed in row. The actuator disk model and full rotor CFD simulations using a variety of turbulence models. Comparisons are carried out amongst the different predictions and against measured velocity profiles. Comments 1. It is mentioned that steady state simulations are carried out in the rotating frame. It is also mentioned that the tunnel geometry is included in the simulations. It is not clear how the two may be combined in full rotor simulations when the wind tunnel cross section is square. Perhaps the utilities of Open-Foam mentioned include the implementation of sliding grids. If this is the case, please specify accordingly the relevant details. 2. One missing information regarding the ADM results

concerns the axial induction. Was a measured power and thrust curve used? Please specify. 3. I assume that inflow turbulence is similarly implemented in both models (ADM and full rotor CFD). The plots in Fig 6 seem to suggest that in the ADM results the level of turbulence is higher upstream of the 1st rotor. Is this correct? And if so is there an explanation? 4. It seems that the ADM model only accounts for thrust and not torque. Please specify. Furthermore, I believe that in Fig 5 the axial flow velocity is recorded and that the full rotor contours correspond to the azimuth averaged velocity. If so, it is important to also compare the axial force. 5. In Fig 6 the two models are compared in terms of k. I noted that the k-e results from the full rotor simulations are not symmetric which implies that they are not averaged in azimuth. If so, then is such a comparison valid? 6. In Fig 8 the ADM results underestimate the acceleration seen in the measurements at both ends of the plot window. This is important when partial wake effects on the loading are of interest. Otherwise I agree that the k-e RNG model out performs amongst the different models. Also in connection to comment #3 it would be very useful if there are thrust or power measurements to make a comparison. 7. As also mentioned in the paper, the slow flow recovery seen in Fig 9 may be related to the evolution of inflow turbulence along the computational domain. Wind tunnel measurements on disks and small rotors indicated that by increasing the TI level faster recovery is obtained. Otherwise, in the specific set up, the k-e realizable model performs rather well. 8. Finally, I would suggest to add in the last section that full rotor simulations should be also checked as regards the evolution of inflow TI.

Conclusion: The contains interesting results, but also some unclear points that need clarification. To my opinion the corresponding revisions are important.

---

## Author Comment (AC1) · 30 Apr 2017

**Reply to the comments of the Anonymous Referee #1.**

Dear reviewer,

Thank you in advance for your valuable time and help.

We are very grateful for your comments.
All the questions are addressed in the next pages.
We took everything into consideration and we have revised the paper following your recommendations.

The following format of answering the questions was chosen:

- Question/Comment (from the reviewer)
- Text from the manuscript (the related text as it was provided in the first manuscript)
- Answer (reply from the authors)
- Changes (new/modified text added to the manuscript)

Additionally, a small paragraph to describe the scope and the added value of the current manuscript from our perspective is given below.

We are at your disposal for any further information and willing to improve further our manuscript by adding all the new plots, figures and bibliography files that are also provided in our reply.

Kind regards,
Nikolaos Stergiannis et al.

"Full HAWT rotor CFD simulations using different RANS turbulence models compared with actuator disk and experimental measurements".

The scope of the current paper is to investigate the performance of different turbulence models to predict the single wake development of a wind turbine rotor under the controlled conditions of a wind tunnel. The advantage of modelling the wake expansion in controlled conditions is that the generated turbulence is only related to the inflow conditions and the rotor geometry. Other parameters such as the atmospheric stratification, complex terrain and varying roughness are not present. To our best knowledge, this is the first paper that addresses the performance of different turbulence RANS models in the wake development by using two different steady state CFD approaches (blade-resolved and ADM) that are validated with experimental measurements in all wake regions (near, mid and far).

**Question 1**
*Page 1, line 21.*
*"don't understand that phrase…"*

Text from the manuscript:
*"Wake predictions are shown to be very sensitive to the choice of the RANS turbulence model. For most cases, the ADM under-predicts the velocity deficit, except for the case of RNG k-ε which showed a superb performance in the mid and far wake. (**line 21**) **The full wind turbine rotor simulations showed good agreement to the experimental data, mainly in the near wake, amplifying the differences between the simplified models.**"*

Answer:
One of the research questions this manuscript aims to answer is how much physics is lost when moving from the advanced blade-resolved simulations to the simplified models. In Figures 8 and 9, the full-rotor simulations predict more details of the flow field at the near wake region (over line L1) and resulted in a "W" shaped pattern like the measurements, whereas the ADM resulted to a "U" shaped wake pattern for the velocity deficit in all wake regions over the three lines. We added the following text in the manuscript to clarify this sentence better.

Changes:
The full wind turbine rotor simulations showed good agreement to the experimental data, mainly in the near wake. They predicted more details of the flow field in contrast with the "U" shaped velocity deficit downstream of the ADM, amplifying the differences between the simplified models in that region.

**Question 2**
*Page 2, line 3.*
*"What wind farm are you talking about; give reference and details. Seems rather big (unless maybe extreme case such as Lillgrund)."*

Text from the manuscript:
*"(line3) **Wake effects can cause total annual power losses up to 30%.**"*

Answer:
Thank you for your comment. Indeed, this value can be observed in extreme cases of large offshore wind farms such as the Lillgrund, Walney, London Array, Nysted and Horns Rev in near neutral stability but over specific wind sectors. Authors have added the references and rephrased the sentence.

Changes:
In full wake conditions the power losses of a downstream wind turbine can easily reach the 40% [1] while in extreme cases observed in very large offshore arrays even the 60% [2]–[4]. Considering the total annual power production of a wind farm, by averaging over all wind sectors, wake effects can cause losses from 8% onshore up to 20% offshore [3].

**Question 3**
*Page 2, line 18.*
*"What do you mean by that? What limitations? Please substantiate."*

Text from the manuscript:
*"Although CFD simulations can be performed on both steady and unsteady state problems, **(line 18) it is common and acceptable to use, with certain limitations, the assumption of steady state approach to resolve a time-varying (unsteady state) problem when one is interested in the mean values of the flow.** The numerical solution of the RANS equations using a two-equation turbulence closure model (Markatos, 1986; Wilcox, 1993) is a common approach with reasonable computational cost for industrial applications involving flows of high Reynolds numbers."*

Answer:
The following text has been added to the manuscript with references.

Changes:
…in the mean values of the flow. The wind turbine is rotating with constant tip-speed-ratio, using fixed yaw and blade pitch angles and the inflow is uniform with small wind speed variations reflected in the turbulence intensity. Therefore, the flow can be treated as steady in the frame of rotation, a common approach in turbomachinery research [5], [6]. The drawback of the steady state frozen-rotor approach, is that the rotating wake will always stay in the same position. The numerical solution of the Reynolds-Averaged Navier Stokes (RANS) equations uses a statistical description of the flow, where flow quantities are split in an average and a fluctuation. To express the Reynolds stresses in terms of mean flow quantities and close the system of equations, several turbulence models have been developed. The use of a two-equation turbulence closure model (Markatos, 1986; Wilcox, 1993) is a common approach, with reasonable computational cost, for industrial applications involving flows of high Reynolds numbers.

**Question 4**
*Eq. 1 and 2.*
*"These are not the correct RANS equations"*

Text from the manuscript:

$$\frac{\partial \rho}{\partial t} + \nabla \cdot (\rho \boldsymbol{u}) = 0 \tag{1}$$

$$\frac{\partial (\rho \boldsymbol{u})}{\partial t} + \boldsymbol{u} \cdot \nabla (\rho \boldsymbol{u}) = -\nabla p + \nabla \cdot \tau + \boldsymbol{S_M} \tag{2}$$

Answer:
Thank you for the comment. Equation 2 has been corrected [7].

Changes:

$$\frac{\partial (\rho \boldsymbol{u})}{\partial t} + \nabla \cdot (\rho \boldsymbol{uu}) = -\nabla p + \nabla \cdot \tau + S_M$$

**Question 5**

*Page 10, line 5.*
*"Really?? And that for a residual reduction of only 4 orders of magnitude??*
*At this number of iterations, why not do DES – these would typically require between 10000*
*and 100000 time steps?"*

Text from the manuscript:
*"(line 5) All the cases converged after approximately 6000 iterations and were run for 10000 iteration. Pressure residuals converged below $10^{-3}$, velocity components below $10^{-5}$ and turbulence variables below $10^{-6}$."*

Answer:
Residuals converged for all cases after 6000 iterations. The residual reduction for pressure was four and in some cases five orders of magnitude which is commonly accepted for industrial applications. Second order numerical schemes were used in all cases. An example of the residuals for the case of full-rotor CFD simulation, using the RNG k-ε turbulence model is given in Figure 1 (please see the appendix) and has been included in the manuscript.
Regarding the question of using DES, the current paper investigates the performance of different RANS turbulence models. Detached Eddy Simulation, couples RANS with Large Eddy Simulations and therefore it is out of the scope of this paper.

Changes:
An example of the residuals for the case of full-rotor CFD simulation, using the RNG k-ε turbulence model is given in Figure 10 of Appendix 2.

**Question 6**

*Page 10, line 5.*
*"What is the computational time, number of processors, possible parallel efficiency, etc."*

Text from the manuscript:
*"(line 5) All the cases converged after approximately 6000 iterations and were run for 10000 iteration. Pressure residuals converged below $10^{-3}$, velocity components below $10^{-5}$ and turbulence variables below $10^{-6}$."*

Answer:
Thank you for your question. The requested details have been added to the manuscript.

Changes:
A total number of 128, 2.2 GHz processors with AMD x86_64 architecture have been used. The computational time was approximately 98 hours for the full-rotor CFD simulations and 52 hours for the ADM and 10000 iterations. The total number of cells per processor was approximately 250.000 cells for the full-rotor and 110.000 cells for the ADM respectively.

**Question 7**

*Page 10, line 10.*
*"Why? No real explanation seems to be given in the paper".*

Text from the manuscript:
*"(line 10) Full-rotor simulations are capable to represent the flow field close to the wind turbine (Fig. 3), but they are over-estimate the velocity deficit at the far wake (Fig. 9)."*

Answer:
An explanation is given in lines 23-27, page 11. The velocity deficit is over-estimated in the far wake region by all the full-rotor CFD simulations. The reason is that there is a severe under-estimation of produced turbulence downstream. This result may be related with the absence of the nacelle and tower. Preliminary full-rotor CFD results, including the nacelle and tower, indicate an improved estimation of the mid and far wake region [Figure 2]. We therefore conclude that under these conditions within the wind tunnel the absence of those two components affects the far wake recovery. It is also observed by other researchers that the increased turbulence intensity downstream is causing a quicker wake recovery due to the lateral mixing of the flow at the wake region. As mentioned in the paper p.18, lines 22-25, this topic needs further investigation and is part of our future work.

**Question 8**

*Page 11, line 16.*
*"What recirculation zone – do you find a recirculation zone near the ADM? Did you correctly implement the model?*

Text from the manuscript:
*"Results of TKE (Fig. 6a) for the ADM case show that the $k-\omega$ and the $k-\omega$ SST turbulence models are less sensitive to the shear stresses caused by the free stream flow at the disk circumference, in contrast with the group of $k-\varepsilon$ turbulence models. (line 16) Instead, we observe that there is enhanced turbulence production at the recirculation zone, behind the disk, where a pressure drop is also present. The wakes of the modified $k-\varepsilon$ turbulence models produce more TKE and vorticity at the blades-tip positions."*

Answer:
Thank you for pointing this out. The world "recirculation" was not correct. There was not any recirculation observed downstream but only deceleration. This sentence has been corrected in the manuscript. Regarding the ADM model, the standard Actuator Disk model which was already implemented in OpenFOAM was used. We have checked the source code and described the model in lines 1-5 of page 6.

Changes:
…is enhanced turbulence production at the low-pressure region, behind the disk…

**Question 9**
*Page 15, line 8.*
*"How so? Can you explain why you expect that? Not clear to me, since shear as well as turbulence levels of at second turbine are different, so also production will be different."*

Text from the manuscript:
*"Since the two rotors are identical, and the TSR of the second wind turbine is adjusted to the expected inflow conditions for optimal operation, **(line 8) we expect equal production of turbulence from the two rotors.** That is also confirmed by the contours of vorticity (Fig. 7b), despite the small difference at the tip-vortexes of the second wind turbine which operates under the wake effect of the upstream rotor."*

Answer:
Thank you for your comments. It is true that the production will be different since the second wind turbine operates in lower wind speed and TSR. This sentence has been removed and corrected.

Changes:
Slightly different structures of vorticity can be observed at the second wind turbine rotor which operates under the wake effects of the upstream rotor. That is also confirmed by…

**Question 10**
*"Why not show experimental results of the second wake?*
*Why also not consider the closer spaced cases?"*

Answer:
We only had available measurements at the single wake, downstream of the first wind turbine. Since the scope of the current paper is to examine the performance of different eddy viscosity models in all the regions covering the near, mid and far wake and validate against experimental data, we used the B3 case. It is the only case in which three different measurements at all the wake regions are available. For the other two cases, only one measurement is available at the near wake downstream.

**Comment 1**
*Section 2.4. Belongs in appendix at best.*

Text from the manuscript:
 *"(section 2.4) Turbulence modelling"*

Answer:
Thank you for your comment. We agree that the subsections of 2.4 should be moved in the appendix following your suggestion. Section 2.4 includes now a small introduction with a reference to the Appendix 1.

Changes:
2.4 Turbulence modelling…tested and compared with measurements. A short description of each turbulence model and the used constants is given in Appendix 1.

**Comment 2**
*Page 5, line 18.*
*"True, but last simulations also do not occur in your blade-resolved simulations (where you use uniform inflow); moreover, they could be accounted for in ADM, e.g., the rotating ADM by the group of Porté-Agel."*

Text from the manuscript:
 *"Once the $U_{ref}$ is known, $C_T$ can be estimated through the thrust curve of the wind turbine generator considered as uniform over the rotor area (Rados et al., 2012; Crasto et al., 2012; Mikkelsen, 2003). **(line 18) However, in operating conditions, the flow across the rotor is very complex with varying span-wise properties, due to the blade characteristics, rotational velocity, turbulence, finite number of blades and also other flow characteristics related to non-uniform inflow conditions, atmospheric boundary layer shear and so on.***
*To overcome the limitations of the standard actuator disk model, several more advanced models like the generalized actuator disk, actuator line model, actuator surface, have been proposed (Mikkelsen, 2003)."*

Answer:
Thank you for your comment. It is true, but the aim of this sentence is to describe briefly the limitations of the standard Actuator Disk model in wind energy applications which was used. Improved simplified models are mentioned in the next sentence. The rotating ADM by the group of Porté-Agel has been included in the manuscript with a reference.

Changes:
*…several more advanced models like the rotating actuator disk model, generalized actuator…*[8]

**Comment 3**
*"Number and order of figures according to occurrence in text."*

Answer:
We could not find any mistakes on the numbering or the order of figures. The numbering is handled by the LaTeX code and we ensured that the Figures are in the correct sections 5.1 (contours) and 5.2 (plots against measurements). Please keep in mind that this is following the manuscript LaTeX format provided by the journal and it is not the finalised paper format.

**Comment 4**
*Page 11, line 12. Referee #1:*
*"I expect the effect of shear stresses only to become apparent more downstream – near the disk, solution will behave more like potential flow."*

Text from the manuscript:
**"(line 12) The iso-surfaces of the Q criterion (Fig. 4), are in agreement with the contours of vorticity (Fig. 7a) in which the strong effect of the shear stresses at the edge of the disk area is apparent."**

Answer:
Shear-stresses will occur at the region of strong velocity gradients which is at the disk circumference. There is not any other source or sink of momentum further downstream. A sentence to clarify this statement has been added to the manuscript.

Changes:
Strong velocity gradients are expected at the disk circumference where the free-stream flow will interact with the boundaries of the disk region of decelerated flow (momentum sink). The iso-surfaces of the Q criterion...

**Comment 5**
*Page 11, line 18.*
*"Sorry but this is really a rather trivial statement"*

Text from the manuscript:
*"Additionally, from the contours of turbulence (Fig. 6a) and of velocity (Fig. 5a) we can conclude that **(line 18) there is a strong connection between the production of turbulence and the wake recovery.**"*

Answer:
The sentence has been rephrased and the impact of turbulence on the wake expansion has been addressed and added to the manuscript.

Changes:
The enhanced turbulence kinetic energy will increase the mixing in between the freestream layers of the flow and the inner wake flow. More enhanced mixing will cause stronger interaction between the regions and finally a faster recovery of the velocity downstream. This mixing will occur more at the far wake downstream, where the impact of the rotor is less strong. The connection between the production of turbulence and of the wake recovery can be observed also in the contours of turbulence (Fig. 6a) and of velocity (Fig. 5a). It is more dominant in the case of k-ω turbulence model, where the over predicted turbulence production was observed with faster mid and far wake velocity recovery.

**Comment 6**
*Page 18.*
*"It is concluded that the k-omega models are the best. Unfortunately, no tower or nacelle is modelled, so not sure that this conclusion will hold one you should add them."*

Answer:
It is true that from the current research outcome for the blade-resolved CFD approach, is that the group of k-ω models performed better. Preliminary results with nacelle and tower suggest that this is always the case, but, as already mentioned in the manuscript, this topic needs further investigation

**Comment 7**
*Page 18, line 17.*
*"Sanderse et al is a review paper and not an ADM study."*

Text from the manuscript:
*"(line 17) The under-prediction of the far wake by ADM was observed also by other studies (Rados et al., 2012; Sanderse et al., 2011; Vafiadis et al., 2013; Crasto et al., 2012; Prospathopoulos et al., 2011)."*

Answer:
Thank you for your comment. The reference to Sanderse et al. has been removed from that part of the manuscript.

**APPENDIX**

Figure 1. Residuals of the full-rotor CFD simulations using the RNG k-ε turbulence model.

[Figure]

Figure 2. Results of the full-rotor CFD simulations including the nacelle and tower.
Top: mid wake (5.18D), bottom: far wake (8.5D).

[Figure]

[Figure]

**References**

[1]     B. Sanderse, S. P. Pijl, and B. Koren, "Review of computational fluid dynamics for wind turbine wake aerodynamics," *Wind Energy*, vol. 14, no. 7, pp. 799–819, Oct. 2011.

[2]     K. Walker *et al.*, "An evaluation of the predictive accuracy of wake effects models for offshore wind farms," *Wind Energy*, vol. 19, no. 5, pp. 979–996, May 2016.

[3]     R. J. Barthelmie *et al.*, "Modelling and measuring flow and wind turbine wakes in large wind farms offshore," *Wind Energy*, vol. 12, no. 5, pp. 431–444, Jul. 2009.

[4]     R. J. Barthelmie *et al.*, "Modelling the impact of wakes on power output at Nysted and Horns Rev."

[5]     E. Van Der Weide, G. Kalitzin, J. Schluter, G. Medic, and D. J. J. Alonso, "On large scale turbomachinery computations," *Cent. Turbul. Res. Annu. Res. Briefs*, 2005.

[6]     D. Graham, H. Branden, J. Moore, and S. D. Connell, "Unsteady vs. steady turbomachinery flow analysis: exploiting large-scale computations to deepen our understanding of turbomachinery flows."

[7]     J. Anderson, E. Dick, G. Degrez, R. Grundmann, J. Degroote, and J. Vierendeels, *Computational Fluid Dynamics, an Introduction*, Third Edit. Springer, 2009.

[8]     Y.-T. Wu and F. Porté-Agel, "Large-Eddy Simulation of Wind-Turbine Wakes: Evaluation of Turbine Parametrisations," *Boundary-Layer Meteorol.*, vol. 138, no. 3, pp. 345–366, 2011.

---

## Author Comment (AC2) · 30 Apr 2017

**Reply to the comments of the Anonymous Referee #2.**

Dear reviewer,

Thank you very much for your positive review and your valuable feedback.

We are very grateful for your comments.
All the unclear points mentioned in the review have been clarified in the next pages.

The following format of answering the questions was chosen:

- Question/Comment (from the reviewer)
- Text from the manuscript (the related text from the manuscript – if needed)
- Answer (reply from the authors)
- Changes (new/modified text added to the manuscript – if needed)

Please do not hesitate to contact us for any further information.
We are at your disposal and willing to improve further our manuscript.

Kind regards,
Nikolaos Stergiannis et al.

**Comment 1**

*"It is mentioned that steady state simulations are carried out in the rotating frame. It is also mentioned that the tunnel geometry is included in the simulations. It is not clear how the two may be combined when the wind tunnel cross section is square. Perhaps the utilities of OpenFOAM mentioned include the implementation of sliding grids. If this is the case please specify accordingly."*

Answer:
Thank you for your comment. This is now clarified in our manuscript. Further figures could be provided and included in the final manuscript if needed.

Changes:
The MRF regions are cylindrical and they are located inside the stationary squared region of the computational domain. The cyclic arbitrary mesh interface (AMI) was used in between the rotating and the non-rotating mesh regions. To limit any possible impact of the interfaces with the flow field, the MRF regions have been extended further downstream. In all the cases under investigation, the rotating frame regions include the rotor-hub geometries and all the wake regions downstream.

**Comment 2**

*"One missing information regarding the ADM results concerns the axial induction. Was a measured power and thrust curve used? Please specify."*

Answer:
Thank you for your comment. The axial induction factor that was used, is based on the measured Ct and Cp. This information is given in line 10, page 6:
*"The coefficients Cp and Ct have been provided from the wind tunnel measurements and were used as inputs to the simplified models."*

**Comment 3**

*"I assume that inflow turbulence is similarly implemented in both models (FR and ADM). The plots in Fig.6 seem to suggest that in the ADM results the level of turbulence is higher upstream of the 1st rotor. Is this correct? And if so is there an explanation?"*

Answer:
We could not see any indication of higher values for the cases of actuator disk models in Fig. 6. The inflow turbulence intensity is indeed similar for the both CFD approaches. We confirm that the same value of 1.98375 $m^2/s^2$ has been used in the inlet for both CFD approaches.

**Comment 4**

*"It seems that the ADM models only accounts for thrust and not torque. Please specify. Furthermore, I believe that in Fig.5 the axial flow velocity is recorded and that the full rotor contours correspond to the azimuth averaged velocity. If so, it is important to also compare the axial force."*

Answer:

The implemented actuator disk model in OpenFOAM uses the induction factor a to simulate the momentum sink in the flow field. The induction factor is calculated based on both the thrust and torque coefficients which are provided by the experimental measurements. While it is true that the induction coefficient is calculated from thrust and torque, the model is purely axial and does not include rotation. The full rotor contours were not averaged over the azimuth. This is a very good point that we have been also considering. It will resolve the drawback of the frozen rotor technique which produces a rotating wake that always stay in the same position. We are developing a post-process function to perform the averaging over the MRF regions for all the flow quantities. Relative results will be added to the final manuscript.

Changes:

…The coefficients Cp and Ct have been provided from the wind tunnel measurements and were used as inputs to the simplified models. While it is true that the induction coefficient is calculated from thrust and torque, the model is purely axial and does not include rotation.

**Comment 5**

*"In Fig 6 the two models are compared in terms of k. I noted that the k-ε results from the full rotor simulations are not symmetric which implies that they are not averaged in azimuth. If so, then is such a comparison valid?"*

Answer:

Thank you for pointing this out. It is similar to the second part of the previous comment. This is a very good point and we are currently working on a custom post-process function that will perform the averaging in azimuth. Contour plots will be updated in the manuscript.

**Comment 6**

*"In Fig.8 the ADM results underestimate the acceleration seen in the measurements at both ends of the plot window. This is important when partial wake effects on the loading are of interest. Otherwise I agree that the k-ε RNG model out performs amongst the different models. Also in comparison to comment #3 it would be very useful if there are thrust or power measurements to make a comparison."*

Answer:

Thank you for your comment. The observed acceleration seen in the measurements at the ends of the plot window is related to the physical presence of the wind tunnel walls. A

developed boundary layer combined with the blockage of the wind turbine will cause a local acceleration at the region between the rotor and the wind tunnel walls. In the CFD simulations the wind tunnel walls are modelled with slip conditions to ensure that all the gradients and the velocity vectors normal to the walls are zero assuming zero surface friction. It should be noted that the underestimation of the acceleration at both ends of the plot window is also observed in the blade resolved simulations since both CFD approaches are using the same computational domain and boundary conditions.

For the case of ADM the thrust and the power are imposed as inputs through their coefficients, therefore, such a comparison would be useful only for the case of the second wind turbine which operates within the wake of the upstream rotor.

**Comment 7**

"As also mentioned in the paper, the slow flow recovery seen in Fig.9 may be related to the evolution of inflow turbulence along the computational domain. Wind tunnel measurements on disks and small rotors indicated that by increasing the TI level faster recovery is obtained. Otherwise, in the specific set up, the k-ε realizable model performs rather well."

Answer:

We agree that the k-ε realizable model performs well at the far wake, but it is considered unreliable from the unphysical results of TKE that observed in the full rotor CFD simulations. Also, in the blade-resolved approach, it is the only model that fails to predict a "W-shaped" near wake velocity deficit. Therefore, we conclude that it's performance is rather good by accident.

**Comment 8**

"Finally, I would suggest to add in the last section that full rotor simulations should be also checked as regards the evolution of inflow TI."

Answer:
True statement, we will add this in the last section.

**Conclusion from the referee 2**

"The paper contains interesting results, but also some unclear points that need clarification. To my opinion, the corresponding revisions are important."

**Reply:** Thank you very much for your positive review. All the unclear points mentioned in the review have been clarified. Please do not hesitate to contact us for any further information.